# Can Intergroup Contact in Virtual Reality (VR) Reduce Stigmatization Against People with Schizophrenia?

**DOI:** 10.3390/jcm10132961

**Published:** 2021-06-30

**Authors:** Daniela Stelzmann, Roland Toth, David Schieferdecker

**Affiliations:** 1Institute of Computer Science, Freie Universität Berlin, 14195 Berlin, Germany; 2Institute for Media and Communication Studies, Freie Universität Berlin, 14195 Berlin, Germany; roland.toth@fu-berlin.de (R.T.); d.schieferdecker@fu-berlin.de (D.S.)

**Keywords:** stigma, schizophrenia, mental disorders, virtual reality (VR), intergroup contact

## Abstract

People with mental disorders such as schizophrenia do not only suffer from the symptoms of their disorders but also from the stigma attached to it. Although direct intergroup contact is an effective tool to reduce stigmatization, it is rare in real life and costly to be established in interventions, and the success of traditional media campaigns is debatable. We propose Virtual Reality (VR) as a low-threshold alternative for establishing contact since it involves less barriers for affected and unaffected persons. In a 2 + 1 experiment (*n* = 114), we compared the effects of encounters with a person with schizophrenia through a VR video with contact through a regular video and no contact at all on anxiety, empathy, social proximity, and benevolence towards people with schizophrenia. We found that contact via VR reduced stigmatization only for participants who liked the person encountered. Our data suggest that it is crucial how participants evaluate the person that they encounter and that stronger perception of spatial presence during reception plays an important role, too. Therefore, we discussvarious boundary conditions that need to be considered in VR interventions and future research on destigmatization towards mental disorders, especially schizophrenia.

## 1. Introduction

In 2017, approximately 792 million people suffered from mental disorders worldwide [1]. Affected persons do not only suffer from the symptoms of their mental disorder, but also from the stigma attached to it [2,3,4]. The consequences of such stigmatization are tremendous: It lowers affected persons’ self-esteem and their likeliness of seeking professional help [5,6,7]. Accordingly, the World Health Organization (WHO) has made stigma reduction one of its top priorities [8].

One approach to reducing stigmatization is to establish meaningful encounters between those who are affected by a mental disorder and those who are not [9]. Although direct intergroup contact is an effective tool for reducing stigmatization in the realm of mental disorders, it only infrequently happens in real life and can be costly to establish in the context of stigmatization campaigns and educational programs. Seeking alternative forms of contact, mass media campaigns yielded non-satisfying results [10], but first studies showed that computer-mediated intergroup contact can be an auspicious tool for decreasing stigmatization [11,12].

We set out to test the potential of a digital technology that has not yet been in focus regarding stigmatization towards people with mental disorders: Virtual Reality (VR). The high sensual richness and involvement make VR a promising tool for intergroup contact [13,14]. In the context of mental disorders, VR technology has mainly been studied in the domain of cognitive behavioral therapy [15]. We are not aware of any study that investigates intergroup contact via VR in the context of mental disorders [9]. To close this gap, we conducted an experiment that tested the effects of VR/360°-3D videos on stigmatization towards people with schizophrenia, one of the most stigmatized mental disorders [16].

### 1.1. Stigma Towards People with Mental Disorders and Their Consequences

According to Goffman, a stigma is a deeply discrediting attribute attached to a person or group [17]. Individuals who suffer from mental disorders are affected by various forms of stigmatization [2,3,4,18]. The symptoms of the mental disorders are often cited as the main cause of stigma since they can lead to deviant behavior and are sometimes hard to comprehend for bystanders [19]. Particularly mental disorders that can induce psychotic episodes—like schizophrenia—are commonly associated with danger, crime, and unpredictability [19,20]. These beliefs neglect the fact that mental disorders such as schizophrenia are neither a necessary, nor a sufficient precondition for violent behavior [21,22,23]. Yet, these stereotypes largely persist because media coverage is perpetuating images of violent “mad men”, serial killers, and psychopaths and involves people with mental disorders like schizophrenia primarily in the context of violence [24,25,26,27].

The stigma around mental disorders such as schizophrenia has real consequences for affected persons. The unaffected majority feels more anxious, less empathic, more socially distant towards affected persons, and is less supportive of policies that benefit people with mental disorders [19,28]. As a result, it can become even harder for affected individuals to engage in positive contact with others—they feel socially isolated and disconnected from society and suffer from low self-esteem [4,29,30,31,32]. Moreover, people with strongly stigmatized mental disorders such as schizophrenia take longer to seek therapy and engage in treatments [5] which may exacerbate existing symptoms [33]. Even when they do engage in therapy, they are often faced with discrimination in the health care system [34].

### 1.2. Reducing Stigma Towards People with Mental Disorders

For a long time, scholars have studied ways to reduce stigmatization towards people with mental disorders [10]. Among others, direct intergroup contact—face-to-face encounters with affected persons—has emerged as an effective tool for improving attitudes towards people with mental disorders in general [9,35] and schizophrenia in particular [36]. Having said this, opportunities for direct contact are scarce in real life because the lifetime prevalence of mental disorders such as schizophrenia is low at approximately 0.5% [37]. Moreover, affected individuals either avoid contact, are being avoided, or will not reveal their condition at all due to fear of stigmatization [38].

In response to the lack of direct contact, researchers investigated indirect forms of contact, among others via media [9]. Contact via media can come as unidirectional, one-sided exposure to one or more out-group member(s) (e.g., an affected person who recounts their medical history). Theoretically, such encounters should be less anxiety-provoking, more scalable for interventions and should protect people with mental disorders from stressful experiences of unsupervised interactions in non-therapeutic settings. Since findings about contact through traditional media are inconsistent and provide no clear effect patterns [10], scholars began exploring contact in digital, computer-mediated settings [39,40]. For instance, Maunder et al. [12] found a reduction of fear, anger, and stereotyping towards people with schizophrenia after computer-mediated interaction.

One technology that has received little attention as a means for intergroup contact and destigmatization, is VR. VR allows recipients to freely adjust their view in a 360° video or a computer-animated space using a headset. Moreover, it enables rendering the perceived scene in 3D as both eyes are provided separate screens. As such, VR allows for high levels of perceived realism and immersion during media reception [14,41].

In the context of mental disorders, VR has been primarily studied in the context of cognitive behavioral therapy for inducing attitudinal and behavioral change in patients with mental disorders, for example, by exposing them to virtual situations that elicited their fears, such as public speaking or spiders [42]. With regards to destigmatization, first research indicates that first-hand experiences of the perceptions of an affected person (e.g., psychotic symptoms such as hearing noises) in a simulated VR environment can increase empathy and positive attitudes towards persons with schizophrenia—at least when the intervention is accompanied with additional empathy-inducing information [43]. Outside the realm of mental disorders, VR was also used to induce empathy by taking virtual first-person perspectives of racialized minorities [44], women experiencing partner violence [45], and elderly help-seeking people [46].

Empirical studies that assess the effects of contact with an out-group member in VR are scarce [14]—and with regards to mental disorder such as schizophrenia, to the best of our knowledge, non-existent. This is a serious shortcoming, since exposure in VR—defined here as one-sided exposure to an affected person—should combine the advantages of direct contact with those of mediated contact. VR should offer levels of sensual richness and involvement—two core dimensions used to categorize different forms of intergroup contact [13]—that are much higher than in other forms of media exposure, even if they lack the interactive quality of direct contact [14]. At the same time, VR contact should not be burdened by the anxiety that is often evoked by face-to-face contact and spares affected people negative experiences, since their physical presence is not required. We therefore ask the following explorative research question:

To what extent can contact with a person with schizophrenia through VR reduce stigma associated with the mental disorder in comparison to no contact at all and a regular video?

## 2. Materials and Methods

We employed a 2 + 1 experimental design with two experimental groups that watched either a VR video or a regular video and a control group.

### 2.1. Stimulus Material

We recorded a video of an actor portraying a young man with schizophrenia using a 360°-3D camera with multiple lenses (*Insta360 Pro*). As we expected our participants to be fairly young, we chose a young actor to increase the likelihood that recipients relate and identify with him due to their age. In line with this, we chose a male actor since males are usually diagnosed with schizophrenia at a younger age compared to females [47]. Sitting on a bench outdoors, the young man spoke of his life with the schizophrenic disorder. In his monologue, he recounted the appearance of first symptoms and his medical history, gave an account of his daily routines, and explained how his loved ones were trying to cope with the situation. Although fictitious, the monologue merged various real-life histories and applied principles of narrative medicine [48] in order to make the person more relatable and to allow for an empathetic response.

We rendered two videos from the recording. The first video was a 360°-3D video that could be watched on a VR device. The option to freely look around the scene and the perception of depth and distance ought to ensure maximum involvement and engagement in the scene. The second video was a regular video with a fixed perspective, centering on the actor. Apart from the rendering modes, the videos were identical.

### 2.2. Participants

Assuming a small effect size, η = 0.15 [43] and allowing for α and β errors of 0.05 each, a priori power analysis suggested a desirable sample size of 120 participants. We recruited 114 participants (n_VR_ = 31, n_Vid_ = 45, n_Con_ = 38) via advertisements at a German public university in June 2019. The mean age was *M* = 24 years (*SD* = 6.6 years), 58% of participants were female and 63% had a high-school diploma. These demographics suggest that our pool of participants consisted of majority, but not solely, students.

### 2.3. Procedure

After arriving at the laboratory, participants were seated in front of a laptop and randomly assigned to one of the three groups. They were informed about the study procedure and their right to stop participation anytime. All participants were informed that they were about to answer questions regarding their perceptions of persons with schizophrenia. Participants in the VR group and in the regular video group were additionally informed that they were about to watch a video of a person who suffered from schizophrenia talking about their experiences, and—in the case of the VR video—that this may cause slight dizziness.

In the VR group, the assistant then revealed the VR headset (*Zeiss VR One* with a *Samsung Galaxy S8*) that was hidden in a drawer and helped the respondent mount it. Participants were then given about 90 seconds to become acquainted with the virtual surroundings. The assistant then started the video on the phone. In the regular video group, the assistant started the video on the laptop. In the control group, no video was shown. All participants then filled out a questionnaire on the laptop. At the end, participants were comprehensibly debriefed and handed over a fact sheet about schizophrenia. For the debriefing document, see the online supplementary material (OSM).

### 2.4. Measures

Since scholars have not agreed upon one unified measure of stigmatization, we used four related constructs as outcome variables: Anxiety, social proximity, empathy, and benevolence. Knowing that intergroup contact is a powerful instrument for altering out-group attitudes [9,35], we controlled for past exposure to people with schizophrenia (i.e., quantity of prior out-group contact). In addition, we controlled for two qualities of the reception experience that are crucial in the context of contact, especially in VR settings: the perceived attraction of the encountered person (i.e., evaluation of the out-group member) and the immersion in the contact situation (i.e., feeling of spatial presence). Finally, we controlled for gender and age as core demographic indicators.

We assessed all measures but contact frequency with a Likert-type scale ranging from 1 (*do not agree at all*) to 7 (*totally agree*). Where applicable, we applied Confirmatory Factor Analysis (CFA) and Exploratory Factor Analysis (EFA) for uncovering underlying dimensions within the scales. For all items and the analysis code, see the OSM.

#### 2.4.1. Anxiety

We measured how anxious a participant would feel meeting a person with schizophrenia using items from intergroup anxiety scales [49,50]. Items included, for example, feeling “nervous” or “alarmed”. The one-dimensional measure consisted of four items (ω = 0.85).

#### 2.4.2. Social Proximity

We measured social proximity to persons with schizophrenia by inverting a scale previously used by Angermeyer and Matschinger [51] and Röhm [52]. Items included, for example, “I would accept a schizophrenic person as a coworker”. The one-dimensional measure contained six items (ω = 0.86).

#### 2.4.3. Empathy

We measured empathy toward persons with schizophrenia using an indicator developed by Kinnebrock et al. [53] and used by Röhm [52]. The scale included items such as “People underestimate the emotional burden caused by schizophrenia”. Since multiple EFA did not lead to a satisfactory solution, we used the two highest-correlated items (*r* = 0.53) and created a mean index (*M* = 5.54, *SD* = 1.06).

#### 2.4.4. Benevolence

We measured benevolence toward persons with schizophrenia with the benevolence dimension of the Community-Attitudes-Toward-the-Mentally-Ill Inventory (CAMI), translated to German by Angermeyer et al. [54]. We replaced the phrase “mentally ill” with “persons with schizophrenia”. Items included, for example: “As a society, we need to adopt a much more tolerant attitude towards persons with schizophrenia.” Since multiple EFA did not lead to a satisfactory solution, we used the two highest-correlated items (*r* = 0.48) and created a mean index (*M* = 6.14, *SD* = 0.99).

#### 2.4.5. Contact

We asked how much contact participants have had with persons with schizophrenia within their “family, circle of friends or acquaintances within the past five years” on a scale ranging from 0 (*not at all*) to 5 (*a lot*).

#### 2.4.6. Spatial Presence

We measured feelings of spatial presence within the VR/regular video with the Spatial Presence sub-scale from the MEC Spatial Presence Questionnaire [55]. It includes items such as “I felt like I was actually there in the environment of the presentation”. The one-dimensional measure included four items (ω = 0.93).

#### 2.4.7. Evaluation

We measured the valence of participants’ evaluation of the person in the VR/regular video with items from the General Evaluation Scale [56] and the intergroup anxiety scale [49]. Attributes included, e.g., “likeable” or “natural”. The one-dimensional measure contained four items (ω = 0.81).

### 2.5. Statistical Analysis

For data cleansing and analysis, we used R (Version 4.0.3; [57]) and the R-packages *lavaan* (Version 0.6.8; [58]), and *tidyverse* (Version 1.3.1; [59]). Additional information, the data, the analysis scripts, and a completely reproducible version of this manuscript can be found in the OSM.

We ran various Structural Equation Models (SEM) to cover comparisons between our experimental groups. We always used anxiety, social proximity, empathy, and benevolence as outcome variables and controlled for the quantity of prior out-group contact, gender, and age. For the comparison between the VR and regular video groups, we additionally controlled for evaluation of the encountered group member and spatial presence during reception. We focus on both statistical significance and effect sizes/directions and report significance at *p* < 0.05.

## 3. Results

See Table 1 for the results.

In Model 1, we compared the VR group with the control group and found that contact in VR did not decrease stigmatization in comparison to not having any exposure at all (*RMSEA* = 0.070, *CFI* = 0.921). None of the effects were significant, and in terms of mere effect sizes, the VR video group showed more social proximity than the control group, but also marginally more anxiety and marginally less empathy and benevolence.

In Model 2, we compared the VR group with the regular video group (*RMSEA* = 0.049, *CFI* = 0.943). In this comparison, VR contact actually increased stigmatization. The VR group showed significantly more anxiety, significantly less social proximity and empathy, and less benevolence judging by the effect only. Contrasting VR with regular video contact allowed us to account for context variables surrounding the reception experience. The evaluation of the group member emerged as a crucial predictor: A more positive evaluation of the encountered person significantly decreased anxiety and significantly increased social proximity and benevolence, as well as empathy, judging by the effect only. Moreover, we found that spatial presence during reception marginally decreased anxiety and increased social proximity, empathy and benevolence—although only the effect on benevolence was significant.

In Model 3, we compared the regular video group with the control group (*RMSEA* = 0.082, *CFI* = 0.873). In comparison to no exposure, the regular video decreased anxiety, increased empathy, social proximity, and marginally benevolence—although only the effect for social proximity reached statistical significance.

To corroborate the impression from Model 2, we re-ran the comparison between the VR group and the control group just for participants who evaluated the group member in the video in a positive way, i.e., who scored higher than the median value, in Model 4 (*RMSEA* = 0.102, *CFI* = 0.821). As a result, the effect of the VR video changed directions. VR exposure significantly increased social proximity and—although not significantly—also increased benevolence and decreased anxiety.

Finally, it is notable that more prior direct contact decreased stigmatization constantly and mostly significantly in all models. Although gender had no significant effect, older participants usually showed higher levels of stigmatization.

## 4. Discussion

This study aimed for contributing to the growing literature on destigmatization of mental disorders via contact interventions by testing the potential of VR technology. Specifically, we investigated whether intergroup contact with a person with schizophrenia through VR technology can decrease stigmatization towards people with schizophrenia.

Our study revealed that contact through VR is not a magic bullet for reducing stigmatization in the context of mental disorders. Encountering a person talking about their life with schizophrenia in VR did not significantly decrease stigmatization. Judging by effect direction only, contact via VR may even have negative effects and perpetuate existing stigma. In comparison to watching the regular video, VR contact was associated with significantly more stigmatization in three out of four outcomes. At the same time, participants who watched the regular video showed slightly less stigma than the control group. At first glance, the negative effects of the VR video intervention are counter-intuitive, since the sensual richness and personal involvement of VR promised to make it an effective form of intergroup contact [14]. However, considering the effects of our covariates, we can offer a first explanation for this finding.

In our study, evaluation of the encountered group member emerged as the most important factor for a successful destigmatization via VR. A positive evaluation was negatively associated with all indicators of stigmatization, regardless of the type of video. Moreover, contact in VR did decrease stigmatization among respondents who evaluated our actor in a positive way. It is therefore essential to establish in advance that the encountered person appears likeable to the target audience. This may be particularly important for mental disorders such as schizophrenia that are stereotypically associated with threat [19,20].

The evaluation of the encountered person was important in VR, but not in the regular video which decreased stigmatization independently of it. Subjects were likely overwhelmed by the VR experience and therefore processed the video in a peripheral, heuristic manner [60]. Among participants in the VR condition, none were routine users of the technology; the majority had rarely tried it and one third reported they had never used it before. For inexperienced users, the location shift and 3D perspective in a virtual environment can be a strong sensual experience [61] — many participants in the VR group may have been primarily concerned with adjusting to it. This probably led to restricted cognitive capacity and paying attention specifically to information in the actor’s monologue that was congruent with existing stereotypes and judging the actor by heuristic cues such as attraction [62]. Moreover, participants may have felt as if they had no control or agency in the contact situation: the setting was a 360°-3D video in which they could not adjust location [63] and had to dismount the headset in order to interrupt the encounter. In such a situation, the level of comfort likely depends more strongly on the likeability of the person encountered. Appropriately, participants who felt higher spatial presence showed slightly less stigma.

Based on our findings, future researchers do not only need to choose a likeable person for VR contact, but simultaneously ensure high levels of immersion without making the experience overwhelming, which is in line with previous research [62]. Several solutions might be feasible. First, participants should be offered even more time to become used to the VR condition, e.g., by showing them one or two unrelated videos in the beginning, and therefore allowing them to later focus on the encounter itself rather than adjusting to the situation [64,65]. Second, once likeable actors are identified, VR-based destigmatization campaigns could repeatedly expose participants to them. Not only could participants grow even more familiar with the technology such as this, but repeated, positive intergroup contact such as friendships are known to be more effective than superficial once-off contact [66]. Familiarity-based liking and decreased (anticipated) anxiety could work particularly well for a group that is conceived as unpredictable. Third, recipients can be part of an immersive setting, yet not actively involved, if the VR video presents a situation of so-called vicarious contact [67], meaning that participants are exposed to an interaction between an out-group and an in-group member rather than directly to the out-group member. Finally, health care professionals—who represent trustworthy sources of health information [68]—could be integrated in the contact setting to moderate, comment and contextualize the narrative medical history.

Altogether, we only expect positive effects from VR encounters if the contact situation is carefully calibrated. The assumed advantages of VR over other forms of mediated contact may turn out to be disadvantages otherwise. Future research should investigate the suggestions introduced above. While it was not in the focus of this article, our results indicate the importance of direct intergroup contact as a means for reducing stigmatization, as the mere quantity of prior contact was constantly associated with less stigmatization.

## 5. Limitations

Our study is a first exploration of the potential of VR contact for reducing stigmatization against persons with schizophrenia. As an explorative study, it is limited regarding four ways. First, using a sample that predominantly consists of students can have advantages for exploratory studies with limited sample size, since age and education are rather homogeneous. However, a more heterogeneous sample would yield more generalizable effects [69]. Second, different mental disorders are associated with different types of stigma. VR interventions might be more effective in the context of mental disorders that are less associated with threat, such as major depressive disorder or eating disorders [19,70,71,72]. Third, while we found several consistent effect patterns among our outcome variables, some of the effects were not statistically significant. We already used a small expected effect size (η = 0.15) in power analysis, so future researchers should use an even smaller one. Finally, we presented participants with a natural setting in which they could not engage in a real conversation with the encountered person, but rather in a para-social form of contact [73]. Since prior studies demonstrated the effectiveness of interactive computer-mediated contact [12], future studies should explore fully animated VR settings that allow for interactions—although these may lack in realism and credibility.

## 6. Conclusions

In contrast to our expectations, contact through VR did not reduce stigmatization as compared to no contact and led to more negative attitudes as compared to contact through a regular video in our study. This effect shifted once we only considered individuals who evaluated the encountered group member positively. Our findings offer first insights into the conditions of successful destigmatization through contact in VR—namely how it could be optimized to maximize positive effects and minimize undesirable outcomes. We hope this study instills a greater engagement with the potential of new technologies for generating pro-social attitudes towards people with mental disorders in general and people with schizophrenia in particular and thus contributes to a better quality of life for those affected.

## Figures and Tables

**Table 1 jcm-10-02961-t001:** SEM results for group difference, spatial presence, evaluation, contact, female gender, and age predicting anxiety, social proximity, empathy, and benevolence.

	Anxiety		Social Proximity		Empathy		Benevolence	
	β	p	β	p	β	p	β	p
Model 1 (VR vs. Control) (*n* = 60)								
VR	−0.009	0.943	0.133	0.281	−0.144	0.234	−0.038	0.750
Contact	−0.426	0.000	0.365	0.003	0.235	0.048	0.214	0.023
Female	0.020	0.867	−0.051	0.691	−0.056	0.640	−0.060	0.608
Age	0.166	0.131	−0.405	0.000	−0.050	0.739	−0.373	0.000
Model 2 (VR vs. Regular video) (*n* = 69)								
VR	0.508	0.032	−0.506	0.039	−0.507	0.046	−0.328	0.196
Spatial presence	−0.052	0.679	0.214	0.097	0.101	0.437	0.247	0.046
Evaluation	−0.323	0.004	0.472	0.000	0.254	0.069	0.363	0.003
Contact	−0.397	0.000	0.378	0.001	0.084	0.493	0.242	0.003
Female	0.080	0.482	−0.032	0.756	−0.003	0.981	0.118	0.232
Age	−0.048	0.547	−0.200	0.023	−0.120	0.124	−0.138	0.403
Model 3 (Regular video vs. Control) (*n* = 74)								
Regular video	−0.175	0.157	0.311	0.020	0.098	0.425	0.005	0.967
Contact	−0.302	0.009	0.271	0.020	0.199	0.101	0.045	0.650
Female	0.014	0.904	−0.029	0.794	−0.113	0.310	0.024	0.833
Age	−0.057	0.507	−0.291	0.011	−0.039	0.718	−0.146	0.400
Model 4 (VR vs. Control, high evaluation) (*n* = 54)								
VR	−0.115	0.425	0.414	0.002	−0.029	0.818	0.223	0.071
Contact	−0.402	0.008	0.298	0.028	0.284	0.052	0.116	0.319
Female	−0.016	0.907	−0.084	0.549	−0.180	0.182	−0.243	0.054
Age	0.121	0.276	−0.467	0.000	0.001	0.994	−0.325	0.001

Note: Robust ML estimation, standardized coefficients.

## Data Availability

All data and code to reproduce the analysis is available at https://osf.io/9m4ex/?view_only=cc93e18a04024103abddaf734dd7507f, accessed 29 June 2021.

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
