# Peer review of "Can Intergroup Contact in Virtual Reality (VR) Reduce Stigmatization Against People with Schizophrenia?"

_jcm, 2021, doi:10.3390/jcm10132961_

Round 1

Reviewer 1 Report

Since stigmatization of mental ilness concerns a serious problem in society, with negative consequences for stigmatized individuals, interventions aimed at decreasing stigmatization are important. In the current study, Virtual Reality (360⁰ videos) was used to enable ‘intergroup contact’. Although no positive effects on stigmatization were found, the authors contemplate on the findings in the discussion and formulate boundary conditions for future studies. The manuscript was well-written and pleasant to read.

Abstract:
In the abstract, the authors suggest that the proposed method is viable (‘We propose contact in virtual reality (VR) as a viable alternative’). I wonder whether it would be viable to buy many VR goggles and instruct people one person at a time. Maybe the authors could look for another characteristic instead of viable? I can imagine it would be viable to, for example, arrange low cost events in communities including both people with a psychotic disorders and people without psychosis. That would enable real contact and two-sided interactions as well.

Introduction:
In the introduction the authors posit that VR technology has mainly been studied in the domain of cognitive-behavioral therapy. ‘However, in the context of mental disorders, VR technology has mainly been studied in the domain of cognitive-behavioral therapy. We are not aware of any study that investigates intergroup contact via VR in the context of mental disorders.’ Although many VR studies focused on CBT, there are also VR experiments that investigated for example the effect of embodiment in someone else’s body on sympathy for this type of person. For example, Banakou et al. 2016. Virtual Embodiment of White People in a Black Virtual Body Leads to a Sustained Reduction in Their Implicit Racial Bias. Or Gonzales-Liencres et al. 2020. Being the Victim of Intimate Partner Violence in Virtual Reality: First- Versus Third-Person Perspective. This may me very relevant for the current study, please look into this and maybe add to the introduction.

Methods:
The research question is formulated as follows ‘To what extent can contact with a person with schizophrenia through VR reduce stigma associated with the mental disorder in comparison to no contact at all and a regular video?’. ‘Contact’ seems to indicate that there is some kind of interaction, while actually participants could only passively look at the actor. The authors may consider adding the lack of interaction as a limitation no of their study, since the paradigm is based on the idea of ‘intergroup contact’.

Discussion:
Compliments for the discussion, the results are critically reviewed and boundary conditions and suggestions for future research are provided.

Author Response

Since stigmatization of mental illness concerns a serious problem in society, with negative consequences for stigmatized individuals, interventions aimed at decreasing stigmatization are important. In the current study, Virtual Reality (360⁰ videos) was used to enable ‘intergroup contact’. Although no positive effects on stigmatization were found, the authors contemplate on the findings in the discussion and formulate boundary conditions for future studies. The manuscript was well-written and pleasant to read.

>> We are grateful for all your thoughts, time and efforts you  invested in our manuscript and are very pleased you recognized the added value of our study. 

Abstract:

In the abstract, the authors suggest that the proposed method is viable (‘We propose contact in virtual reality (VR) as a viable alternative’). I wonder whether it would be viable to buy many VR goggles and instruct people one person at a time. Maybe the authors could look for another characteristic instead of viable? I can imagine it would be viable to, for example, arrange low cost events in communities including both people with a psychotic disorders and people without psychosis. That would enable real contact and two-sided interactions as well.

>> We thank you for this point. We agree that the term “viable” may be misleading for our readers in this context and changed the word. From our perspective, indirect contact via VR is primarily a low-threshold alternative as it is associated with fewer barriers than direct contact in analog settings. 

Therefore, we adjusted the sentence into “We propose Virtual Reality (VR) as a low-threshold alternative for establishing contact since it involves less barriers for affected and unaffected persons”

Introduction:

In the introduction the authors posit that VR technology has mainly been studied in the domain of cognitive-behavioral therapy. ‘However, in the context of mental disorders, VR technology has mainly been studied in the domain of cognitive-behavioral therapy. We are not aware of any study that investigates intergroup contact via VR in the context of mental disorders.’ Although many VR studies focused on CBT, there are also VR experiments that investigated for example the effect of embodiment in someone else’s body on sympathy for this type of person. For example, Banakou et al. 2016. Virtual Embodiment of White People in a Black Virtual Body Leads to a Sustained Reduction in Their Implicit Racial Bias. Or Gonzales-Liencres et al. 2020. Being the Victim of Intimate Partner Violence in Virtual Reality: First- Versus Third-Person Perspective. This may me very relevant for the current study, please look into this and maybe add to the introduction.

>> We thank you for this remark. We revised the respective paragraph and now point towards the existing literature that studied VR as a means for perspective-taking and empathy-induction. We happily included both references you suggested along with further research. We adjusted the section as follows: “Outside the realm of mental disorders, VR was also used to induce empathy by taking virtual first-person perspectives of racialized minorities [44], women experiencing partner violence [45], and elderly help-seeking people [46].”

Methods:

The research question is formulated as follows ‘To what extent can contact with a person with schizophrenia through VR reduce stigma associated with the mental disorder in comparison to no contact at all and a regular video?’. ‘Contact’ seems to indicate that there is some kind of interaction, while actually participants could only passively look at the actor. The authors may consider adding the lack of interaction as a limitation no of their study, since the paradigm is based on the idea of ‘intergroup contact’.

>> We thank you  for this sharp observation and an important point. The current intergroup contact literature treats various forms of exposure as contact (ranging from imaging an encounter, over observing an interaction between in- and outgroup members, to long-term friendships between in- and outgroup members). We agree  that with regard to VR contact lacking a fundamental quality of direct contact, the interactivity, albeit VR is more immersive than contact/exposure in other media. In the revised document, we are more explicit about this point and rephrased the respective phrases in the literature review. Moreover, acknowledging the importance of this point, we added it as an additional limitation that future studies need to address: “Finally, we presented participants with a natural setting in which they could not engage in a real conversation with the encountered person, but rather in a para-social form of contact [73]. Since prior studies demonstrated the effectiveness of interactive computer-mediated contact [12], future studies should explore fully-animated VR settings that allow for interactions – although these may lack in realism and credibility.”

Discussion:

Compliments for the discussion, the results are critically reviewed and boundary conditions and suggestions for future research are provided.

>> We thank you for the feedback on the discussion.

Reviewer 2 Report

Dear Authors, 

Thank you for the opportunity to review the paper entitled “Can Contact in Virtual Reality Reduce Stigmatization Against People with Schizophrenia?”. The study compared the effects of encounters with a person with schizophrenia through a VR video with contact through a regular video and no contact at all on anxiety, empathy, social proximity, and benevolence towards people with schizophrenia. The paper presents a very interesting field of study and has a clear purpose. I would like to congratulate the authors for their very interesting study. The design of the paper, and the methodology of the study are correct. Appropriate data analysis methods were used, and the results were discussed properly. I believe the study brings an important practical message. However, I have a major problem with the language/style of the manuscript. In my opinion it is not very scientific, more literary/humanistic. I don't have any major concerns except for the language. Below is a list of my comments:

Minor:

  • Introduction, don’t understand the meaning of “[overview, see:..].”, “see a recent meta-analysis…”,”[for an overview, see…” - I don't know if this is in line with editorial requirements, but it's the first time I've seen such links
  • Who were the participants in the study ?
  • I miss section 2.5 Statistical Analysis, where the information from the beginning of the results should be placed

Author Response

Dear Authors, 

Thank you for the opportunity to review the paper entitled “Can Contact in Virtual Reality Reduce Stigmatization Against People with Schizophrenia?”. The study compared the effects of encounters with a person with schizophrenia through a VR video with contact through a regular video and no contact at all on anxiety, empathy, social proximity, and benevolence towards people with schizophrenia. The paper presents a very interesting field of study and has a clear purpose. I would like to congratulate the authors for their very interesting study. The design of the paper, and the methodology of the study are correct. Appropriate data analysis methods were used, and the results were discussed properly. I believe the study brings an important practical message.

>> We thank you for recognizing the contribution of our study to the field of research and the methodological approach.

However, I have a major problem with the language/style of the manuscript. In my opinion it is not very scientific, more literary/humanistic. I don't have any major concerns except for the language.

>> We thank you for making us aware that the language/style of the manuscript sounded too literary/humanistic. We took your comment very seriously and revised the whole manuscript regarding style - we deleted informal expressions and unnecessary stylistic devices (e.g., “best of both worlds”). We are very thankful for your remark, because we naturally want to avoid any scepticism towards our findings that is solely based on stylistic conventions. 

Introduction, don’t understand the meaning of “[overview, see:..].”, “see a recent meta-analysis…”,”[for an overview, see…” - I don't know if this is in line with editorial requirements, but it's the first time I've seen such links

>> We thank you for this comment. We commonly use these links to highlight instances in which a citation refers to a systematic review or meta-analysis. In order to avoid any confusion, we deleted all instances of these links. 

Who were the participants in the study ?

>> In addition to the descriptives of the socio-demographic profile of our sample (gender, age, educational background), we added a sentence that explicitly points out that our participants were in majority, but not solely, students. Otherwise, we are not entirely sure as to what type of information was missing from your perspective.

I miss section 2.5 Statistical Analysis, where the information from the beginning of the results should be placed

>> Many thanks for this comment. In response to your comment, we now report the technicalities of our analysis in a separate sub-section.